# Mechanistic Study of Novel Dipeptidyl Peptidase IV Inhibitory Peptides from Goat’s Milk Based on Peptidomics and In Silico Analysis

**DOI:** 10.3390/foods13081194

**Published:** 2024-04-14

**Authors:** Yulong Wu, Jin Zhang, Ruikai Zhu, Hong Zhang, Dapeng Li, Huanhuan Li, Honggang Tang, Lihong Chen, Xinyan Peng, Xianrong Xu, Ke Zhao

**Affiliations:** 1School of Public Health, Hangzhou Normal University, Hangzhou 311121, China; yulongwu212@163.com (Y.W.); 15924355231@163.com (R.Z.); 2Institute of Food Science, Zhejiang Academy of Agricultural Sciences, Hangzhou 310021, China; lelouchok@126.com (J.Z.); moxq@snnu.edu.cn (H.Z.); ldpyantaidaxue@163.com (D.L.); huanhuanlee325@126.com (H.L.); zaastang@163.com (H.T.); cwc528@163.com (L.C.); 3College of Food Engineering and Nutritional Science, Shaanxi Normal University, Xi’an 710062, China; 4College of Life Science, Yantai University, Yantai 264005, China; pengxinyan2006@ytu.edu.cn

**Keywords:** dipeptidyl peptidase IV, inhibitory peptide, goat milk, LC-MS/MS, virtual screening, molecular docking

## Abstract

Two novel dipeptidyl peptidase IV (DPP-IV) inhibitory peptides (YPF and LLLP) were discovered from goat milk protein by peptidomics, in silico analysis, and in vitro assessment. A total of 698 peptides (<23 AA) were successfully identified by LC-MS/MS from goat milk hydrolysates (hydrolyzed by papaian plus proteinase K). Then, 105 potential DPP-IV inhibitory peptides were screened using PeptideRanker, the ToxinPred tool, Libdock, iDPPIV-SCM, and sequence characteristics. After ADME, physicochemical property evaluation, and a literature search, 12 candidates were efficiently selected and synthesized in vitro for functional validation. Two peptides (YPF and LLLP) were found to exert relatively high in vitro chemical system (IC_50_ = 368.54 ± 12.97 μM and 213.99 ± 0.64 μM) and in situ (IC_50_ = 159.46 ± 17.40 μM and 154.96 ± 8.41 μM) DPP-IV inhibitory capacities, and their inhibitory mechanisms were further explored by molecular docking. Our study showed that the formation of strong non-bonding interactions with the core residues from the pocket of DPP-IV (such as ARG358, PHE357, GLU205, TYR662, TYR547, and TYR666) might primarily account for the DPP-IV inhibitory activity of two identified peptides. Overall, the two novel DPP-IV inhibitory peptides rapidly identified in this study can be used as functional food ingredients for the control of diabetes.

## 1. Introduction

Type 2 diabetes mellitus (T2DM) is a prevalent chronic metabolic disease that affect approximately 463 million individuals in 2019 worldwide, and this number will reach 700 million by 2045 [1,2]. Several glucose-lowering medications have been developed for the treatment of T2DM, while dipeptidyl peptidase IV (DPP-IV) inhibitors have recently attracted more attention [3]. DPP-IV is a homodimeric serine protease that plays an important role in maintaining glucose homeostasis by degrading endogenous incretin hormones including glucose-dependent insulinotropic polypeptide (GIP), glucagon-like peptide 1 (GLP-1), and polypeptide YY (PYY) [4]. Therefore, DPP-IV inhibitors, such as linagliptin, sitagliptin, and saxagliptin, are commonly prescribed by physicians [5] and have been demonstrated to be safe and effective in the treatment of T2DM [3]. However, several studies also reported some adverse reactions to these synthetic agents, such as gastrointestinal problems [6], infections [7], immune dysfunction, and skin reactions [8]. Additionally, the high cost of these drugs exposes a significant economic burden [9] and leads to the discontinuation of treatment [1]. These challenges highlight the necessity to find new treatments for preventing, managing, and treating diabetes.

Some food-protein-derived bioactive peptides with DPP-IV inhibitory activity have been identified in several studies [10,11,12,13,14,15,16]. In vitro studies indicated that these peptides showed highly competitive/non-competitive mixed-type inhibition against the DPP-IV enzyme, with an inhibition rate ranging from 45% [14] to 83.81% [16]. Studies from diabetic animal models also suggested that administration of these peptides could reduce glycemia and improve insulin secretion and sensitivity [15,17,18]. However, the identification, enrichment in food, and difficulties in the preparation of these peptides may limit their clinical application and industrial production.

Goat milk is widely consumed around the world for its extensive nutritional properties that make it exceptional and maintain the health of children and adults [19]. Bioactive peptides derived from goat milk exert multifunctional properties, including anti-microbial, immunomodulatory, cholesterol-lowering, anti-oxidant, anti-thrombotic, antagonistic activities against various toxic agents [20], as well as the inhibitory activities on DPP-IV [21]. Novel DPP-IV inhibitory peptides have continuously been identified by different studies [21,22], highlighting the immense value of goat milk in producing the food-protein-derived DPP-IV inhibitor. It has been reported that compared with cow milk, goat milk has higher digestibility, lower sensitization, and health-promoting benefits [23]. However, the traditional methods of identifying these peptides are mainly based on stepwise isolation by a multidimensional column chromatograph, and sequencing by UPLC-MS/MS, which have been proven to be inefficient, time-consuming, and labor-intensive, relying heavily on advanced instruments and trained personnel [24]. Thus, new techniques, such as peptidomics and virtual screening, have been developed and applied to the rapid screening and identification of these bioactive peptides [24,25]. The studies conducted on Sichuan pepper seed hydrolysate [26], whey microbial decomposition [27], and shrimp byproducts autolysate [28] all suggested that the combined application of peptidomics and bioinformatics, a tool widely used in protein structure viewing, analysis, prediction, biochemical feature calculation, and so on [29], is effective and feasible in the identification of food-delivered bioactive peptides.

Therefore, the objectives of this study were to rapidly identify and screen potential DPP-IV inhibitory peptides from goat milk based on in silico screening, LC-MS/MS, molecular docking, and in vitro assessment. The sequence characteristics and the molecular binding mechanisms of these peptides were also investigated. This study may contribute to the rapid screening of DPP-IV inhibitory peptides from goat milk and promote the application of goat milk as functional food.

## 2. Materials and Methods

### 2.1. Materials

Saanen goat milk from Weinan City, China, which contained 3.6% protein, 5.8% fat, 4.6% lactose, and 0.86% minerals, was selected as the experimental milk. Goat milk collected between September and October 2021 was transported by ice packs under aseptic conditions and then stored at −80 °C until use. Proteinase K (≥30 units/mg protein) and papain (≥2000 units/mg) were purchased from Aladdin Biochemical Technology Co., Ltd. (Shanghai, China). Sodium acetate buffer, Tris-HCl buffer, and diprotin A (IPI) were bought from Yuanye Bio-technology Co., Ltd. (Shanghai, China). DPP-IV (EC: 3.4.14.5, human, ≥0.01 mg/mL) and glycyl-prolyl-p-nitroanilide hydrochloride (Gly-Pro-pNA, ≥99%) were obtained from Sigma-Aldrich (St. Louis, MO, USA). Caco-2 cells were bought from the Cell bank of Chinese Academy of Sciences (Shanghai, China). Fetal bovine serum (FBS) was purchased from HyClone (Logan, UT, USA). Dulbecco’s modified Eagle medium (DMEM) was obtained from Gibco Co., Ltd. (Carlsbad, CA, USA). Penicillin–streptomycin liquid, 0.25% trypsin−ethylene diamine tetraacetic acid (EDTA), and phosphate-buffered solution (PBS) were obtained from Solarbio Science & Technology Co., Ltd. (Beijing, China). 3-(4,5-dimethylthiazol-2-thiazolyl)-2,5-diphenyl-2H-tetrazolium bromide (MTT) was bought from Labgic Technology Co., Ltd. (Beijing, China). Gly-Pro-AMC was purchased from AAT Bioquest Inc. (Sunnyvale, CA, USA). All the chemicals and reagents were of analytical grade. Discovery Studio version 2019 (DS2019, Accelrys, Inc., San Diego, CA, USA) was used for virtual screening and molecular docking analysis. All synthesized peptides (purity > 95%) were obtained from China Peptides Co., Ltd. (Shanghai, China).

### 2.2. In Silico-Simulated Hydrolysis of DPP-IV Inhibitory Peptides

Simulated hydrolysis processes for the release of DPP-IV inhibitory peptides from goat milk protein were performed according to the method described by Lin et al. [30] with slight modifications. Briefly, sequences of six major goat milk proteins were downloaded from UniProt Knowledgebase (http://www.uniprot.org/, accessed on 20 November 2021), alpha-S1-casein (P18626), alpha-S2-casein (P33049), beta-casein (P33048), kappa-casein (P02670), alpha-lactalbumin (P00712), and beta-lactoglobulin (P02756), and then subjected to in silico proteolysis. To obtain DPP-IV inhibitory peptide fragments from goat casein and whey protein, five enzymes commonly used in the food industry (proteinase K, thermolysin, papain, stem bromelain, and subtilisin) were applied individually or in combination (two- or three-enzyme combinations) for in silico proteolysis using the “ENZYME(S) ACTION” tool of the BIOPEP-UWM database (https://biochemia.uwm.edu.pl/biopep-uwm/, accessed on 9 December 2021). The ∑A_DPP-IV_ inhibition values of the different proteolysis combinations were recorded and calculated.

### 2.3. Enzymatic Hydrolysis of Goat Milk Protein

Based on the results of the in silico-simulated hydrolysis above, the enzymes that had the highest ∑ADPP-IV inhibition values among all the 15 groups were used for further analysis (proteinase K and papain, as shown in Appendix A). Then, the goat casein and whey proteins were hydrolyzed by the two-step method as follows: goat casein and whey protein were dissolved in deionized water to a concentration of 5% (*v*/*w*) and 25% (*v*/*w*), respectively; then, the proteinase K was first added to the protein solution (55 °C, pH 8, 120 min), followed by the addition of papain (55 °C, pH 7, 90 min and 60 min, respectively) according to the manufacturer’s instructions. After enzyme inactivation at 90 °C and appropriate cooling, the crude casein and whey protein hydrolysate were collected for hyper-filtration using 3 KDa ultrafiltration devices (Merck Millipore, St. Louis, MO, USA). The solutions with a molecular weight lower than 3 KDa were freeze-dried and stored at −80 for further analysis.

### 2.4. Peptidomic Profiling Using LC-MS/MS

The freeze-dried hydrolysate was first desalted by C18 Cartridges (Empore, SPE Cartridges C18, 7 mm inner diameter, 3 mL volumes, Sigma, St. Louis, MO, USA) and vacuum-evaporated. Then, the samples were reconstituted in 40 μL of 0.1% TFA solution and analyzed by LC-MS/MS using Q mass spectrometry coupled to Easy nLC (Thermo Fisher Scientific, Waltham, MA, USA). Briefly, the peptide mixtures were loaded onto a C18-reversed-phase column (25 cm × 75 μm id, Thermo Fisher Scientific, Waltham, MA, USA), packed in-house with RP-C18 5 μm resin in buffer A (0.1% formic acid), and separated with a linear gradient of buffer B (0.1% formic acid in 84% acetonitrile) at a flow rate of 250 nL/min over 60 min. Based on extracted ion chromatograms and spectral counts, validation label-free peptide quantification was performed in the MaxQuant software (version 1.5.5.1). The processed sequences were searched against the NCBI_*Capra*_*hircus*_52489_20221102 (containing 52,489 sequences, downloaded on 2 November 2022) without specifying enzyme cleavage rules. The search parameters were set as follows: ±20 ppm for peptide mass tolerance, 0.1 Da for MS/MS tolerance, and 2 for maximum missed cleavage (with an allowance for 2 missed cleavages). Variable modification: oxidation (M). The cutoff value for the global false discovery rate (FDR) for peptide identification was set to 0.01.

### 2.5. Screening of DPP-IV Inhibitory Peptides

#### 2.5.1. Virtual Screening of Potential DPP-IV Inhibitory Peptides

First, PeptideRanker (http://distilldeep.ucd.ie/PeptideRanker/, accessed on 3 March 2023) was used for the initial screening of bioactive peptides and the toxicity was evaluated by ToxinPred (http://crdd.osdd.net/raghava/toxinpred/, accessed on 3 March 2023) with an SVM threshold value of 0. Then, peptides with predicting scores above 0.5 in PeptideRanker and without toxicity were regarded as bioactive peptides, which were used for further virtual screening of potential DPP-IV inhibitory peptides using iDPPIV-SCM (http://camt.pythonanywhere.com/iDPPIV-SCM, accessed on 3 March 2023), molecular docking, and DPP-IV sequence characteristics. Peptides with a score greater than 294 by iDPPIV-SCM prediction were considered as potential DPP-IV inhibitory peptides. In addition, Discovery Studio 2019 (DS)/the LibDock protocol with default parameters was applied to conduct molecular docking [24]. After pretreating with the DS/Prepare protein protocol, the binding site (x: 39.244430, y: −7.918825, z: 94.030481, and radius of 9 Å) of the DPP-IV protein complex with HL1 (PDB ID: 5J3J) [31] was used as the docking of those potential bioactive peptides, and the peptides with docking success were considered to exert potential DPP-IV inhibition activity. Finally, peptides whose amino acid residues at the N-terminal were Trp/Leu/Ile/Phe, and/or Pro/Ala at the 2nd N-terminal, and/or Pro at the C-terminal were selected, as these were considered characteristic of DPP-IV inhibitory peptides [32]. Overall, the peptides that passed all the three screening methods were screened as target peptides and used for further analysis.

#### 2.5.2. In Silico Analysis of Pharmacokinetic Properties and Physicochemical Properties

The pharmacokinetic parameters ADME (absorption, distribution, metabolism, and excretion) and physicochemical parameters of those potential DPP-IV inhibitory peptides mentioned above were analyzed by SwissADME (http://www.swissadme.ch/, accessed on 6 March 2023). The Simplified Molecular Input Line Entry Specification (SMILES) strings of the selected peptides were generated using ChemDraw 20.0 software. Compared with reference Diprotin A (IPI), the peptides that had at least water solubility and high gastrointestinal (GI) absorption and were not inhibitors of P450 (CYP) (five major isoforms: CYP1A2, CYP2C19, CYP2C9, CYP2D6, and CYP3A4) attracted our attention, as they were regarded as practical in application on drug-like biomolecules [31,33,34]. Then, the screened peptides were analyzed using BIOPEP (https://biochemia.uwm.edu.pl/biopep/start_biopep.php, accessed on 15 March 2023), MBPDB (http://mbpdb.nws.oregonstate.edu/, accessed on 15 March 2023), as well as web of science (https://www.webofscience.com/wos/, accessed on 15 March 2023) and PUMED (https://pubmed.ncbi.nlm.nih.gov/, accessed on 15 March 2023) to find unreported DPP-IV inhibitory peptides.

### 2.6. Solid-Phase Synthesis of Peptides

The selected DPP-IV inhibitory peptides were synthesized by a solid-phase procedure (China Peptides Co., Ltd., Shanghai, China) and the purity was validated as >95%.

### 2.7. DPP-IV Inhibitory Assay

The DPP-IV inhibitory activity of peptides was performed as previously reported with slight modifications [21]. Briefly, peptides were diluted using 100 mM Tris-HCL buffer (pH = 8). Then, 25 μL diluted peptides were mixed with 25 μL of 1.6 mM Gly-Pro-pNA in a 96-well microplate and incubated at 37 °C for 10 min. Afterward, the mixture was added with 50 μL of 315 μg/L DPP-IV solution and reacted at 37 °C for 60 min. Finally, 100 μL of 1 M sodium acetate buffer (pH = 4) was added to stop the reaction, and the absorbance was measured at 405 nm using a SpectraMax 190 microplate reader (Molecular Devices, San Jose, CA, USA). Diprotin A (IPI), a well-known DPP-IV inhibitory peptide [35], was used as a positive control. The DPP-IV-inhibition (DI) was calculated with the following equation:$$DI\left(\%\right)=\left[1-\left(\frac{A_{S}-A_{SC}}{A_{NR}-A_{NC}}\right)\right]\times 100\%$$
where *A_S_* (sample) is the optical density in the presence of peptide, DPP-IV, and Gly-Pro-pNA; *A_SC_* (sample control) is the optical density when DPP-IV is replaced with Tris-HCl buffer (100 mM, pH 8.0); *A_NR_* (negative reaction) is the optical density in the presence of only DPP-IV solution and Gly-Pro-pNA; and *A_NC_* (negative control) is the optical density in the absence of DPP-IV and peptide.

### 2.8. In Situ Cell-Based DPP-IV Activity Assay

#### 2.8.1. Cell Culture

The Caco-2 cells (Cell Bank of Shanghai Academy of Biological Science, Shanghai, China) were incubated at 37 °C in a 5% CO_2_ environment in DMEM supplemented with 10% FBS and 1% penicillin–streptomycin liquid. Unless otherwise stated below, the cells were cultivated at a density of 5 × 10^4^ cells/mL in the culture dishes (Corning Inc., Corning, NY, USA). The cell culture medium was changed every two days, and the cells from passage numbers of 15–25 were used in this study.

#### 2.8.2. Cell Viability Assay

The cell viability was determined following the method described by You et al. [36] with slight modification. Briefly, Caco-2 cells were seeded in 96-well microplates with a density of 1 × 10^4^ cells/well and incubated at 37 °C with 5% CO_2_ for 24 h. Then, different concentrations of YPF or LLLP (0, 0.025, 0.05, 0.1, 0.25, 0.5, and 1 mg/mL) were added to the cells. After 12 h of intervention, 10 μL of MTT (0.05 mg) was added and incubated for another 4 h. Finally, the DMEM medium was replaced with 100 μL of DMSO and the absorbance of the solubilized crystals was detected at 490 nm (Molecular Devices SpectraMax M5, San Jose, CA, USA). The cell viability was calculated by the following equation:Cell viability (%) = (OD_test_ − OD_blank_)/(OD_control_ − OD_blank_) × 100%

#### 2.8.3. In Situ Inhibition of DPP-IV

The inhibitory capacity of YPF or LLLP on DPP-IV was assayed following the method of You et al. [36] with slight modifications. Briefly, 2 × 10^4^ Caco-2 cells/well cells were seeded in 96-well microplates and cultivated for 24 h. The DMEM medium was replaced by 100 μL of YPF or LLLP (0.025, 0.05, 0.1, 0.25, 0.5, and 1 mg/mL) or PBS (100 mM, pH 8.0) for 1 h at 37 °C. Subsequently, 100 µL of the fluorescent substrate Gly-Pro-AMC (50 µM) was added to each well and incubated for 5 min. IPI was used as a positive control. Then, the florescence was monitored with excitation and emission wavelengths of 380 nm and 500 nm (Molecular Devices SpectraMax M5, San Jose, CA, USA).

### 2.9. Molecular Docking Analysis

Molecular docking is usually used to predict the possible binding ability between ligands and receptors. It was reported that the higher the binding capacity of the ligand to enzymes, the lower its binding energy [24]. The structures of all peptides were energetically optimized using the CHARMm force field by the DS/Minimize Ligands protocol, while human DPP-IV in complex with HL1 (PDB ID: 5J3J) was pretreated with the DS/Prepare protein protocol, including cleaning proteins (standardize names, complete residues, hydrogen modification, atom ordering, etc.), inserting missing loops, refining loops, minimizing loops, and protonating proteins. Molecular docking was performed using the CDOCKER protocol in DS2019 and evaluated based on the -CDOCKER interaction energy (-CIE) of the docking results for peptide-5J3J interaction [37]. To evaluate the potential mechanism of DPP-IV inhibitory peptides, the binding sites and non-bonding interaction patterns of the peptides with 5J3J were further analyzed.

### 2.10. Statistical Analysis

Results are shown as means ± standard deviation (SD) of at least three independent experiments. Origin V2022 software (OriginLab, Northampton, NC, USA) and GraphPad Prism 9.5.1 (San Diego, CA, USA) were applied for the data analysis, and the data on the ∑ADPP-IV inhibition value and DPP-IV inhibition were analyzed using one-way ANOVA followed by Dunnett’s multiple comparisons post-test. Statistical significance was set at *p* < 0.05.

## 3. Results and Discussion

### 3.1. Analysis of Goat Milk Protein Hydrolysate Enriched in DPP-IV Inhibitory Peptides

As shown in Appendix A, the most effective combination for the production of the DPP-IV inhibitory peptide in silico was proteinase K plus papain, because its hydrolysis result ∑ADPP-IV had the highest inhibitory value of 0.7622. Then, the casein and whey proteins extracted from fresh goat milk were enzymolized by the two enzymes. A total of 698 peptides detected by LC-MS/MS analysis from casein and whey hydrolysates with peptide identification scores (PISs) >0 are summarized in Figure 1, and the basepeak plots obtained are shown in Appendix A. Among them, 349 and 594 peptides originated from casein and whey protein, respectively, while 245 peptides were commonly detected (Appendix A). Interestingly, 638 peptides were found to have 3–5 amino acid residuals (201 tripeptides, 426 tetrapeptides, and 11 pentapeptides), which accounted for 91.4% of all the detected peptides (Figure 1B). Most peptides of 3–5 amino acid residues have been reported to have DPP-IV inhibitory activity [38], because short peptides that act as competitive inhibitors of DPP-IV readily access the active site of the DPP-IV enzyme for inhibition [39]. Three competitive DPP-IV inhibitory peptides, VPV (IC_50_ = 6.6 μM), YPI (IC_50_ = 35.0 μM), and VPF (IC_50_ = 55.1 μM), have been reported from camel whey protein [40]. In addition, the molecular weight of 541 peptides was less than 500 Dalton, accounting for 77.5% of the total detected peptides (Figure 1C). It was reported that the activities of DPP-IV inhibitory peptides were negatively correlated with their molecular weights [24], and short peptides with low molecular weight were important characteristics of DPP-IV inhibitory peptides [41]. Furthermore, amino acid residues, such as Trp/Leu/Ile/Phe at the N-terminus, and/or Pro/Ala at the second N-terminus, and/or Pro at the C-terminus in the sequence of peptides, represented the characteristics of DPP-IV inhibitory peptides [32], and the total frequency of amino acid residuals at those positions was 41.55% in the 698 identified peptides (Figure 1D). Overall, the results above indicated that the goat milk protein hydrolysate enzymolized by proteinase K plus papain potentially contained large amounts of DPP-IV inhibitory peptides. It was shown that the amino acid sequences of goat and bovine β-casein have a high degree of homology (91%) [42], so it could be assumed that goat milk was homologous to cow milk protein. In addition, homologous proteins have the potential to produce peptides with similar biological functions [30], so cow milk also has the potential to produce large amounts of DPP-IV inhibitory peptides.

### 3.2. The Amino Acid Profiles of Identified Peptides

It was shown that peptides containing hydrophobic amino acids at the N-terminal could increase the specificity of DPP-IV substrates [43], and that the DPP-IV inhibitory ability of peptides correlates with their primary sequence. Considering the potential role of peptide length in DPP-IV inhibitory activities, 698 peptides were categorized into four groups—698 total identified peptides (Figure 2A), tripeptides (Figure 2B), tetrapeptides (Figure 2C), and other peptides with more than four amino acid residuals (Figure 2D)—and the frequency of specific amino acid residuals in the N-terminal, the second N-terminal, and the C-terminal of peptides was further investigated in each group. The results showed that residue Val was the predominant residue at the N-terminal (about 22.35%) and residue Leu was the predominant residue at the C-terminal (about 21.35%) and second N-terminal (17.05%) of the 698 total identified peptides (Figure 2A). These results are consistent with those of Mu et al. [44], who found that hydrophobic amino acids, including Leu, Iie, and Val, accounted for a large percentage of the N-terminal amino acid of DPP-IV inhibitory peptides. Because of those characteristics mentioned above, the peptides could be easily cleaved by DPP-IV [45] and competitively bind to the active sites of DPP-IV [46].

The amino acid profiles of the peptides with different lengths showed a distinct pattern. The Tyr, Leu, and Phe were predominant residues at the N-terminal, the second N-terminal, and the C-terminal of tripeptides identified in our study, respectively (Figure 2B). Residues Phe, Gln, and Lys showed larger frequency in the C-terminal than that in the N-terminal and second N-terminal of tripeptides. However, residues Gly, Ile, and Cys were not found in these peptides. In addition, the high proportion of Leu (12.44%), Phe (9.95%), and Trp (14.43%) at the N-terminus of the tripeptide in Figure 2B indicated the presence of a large number of potential DPP-IV inhibitory peptides. On the other hand, the amino acid profiles of the peptides with >3 residues were significantly different (Figure 2C,D). Although no obvious rule was shown in their amino acid distribution, residues Val and Leu accounted for 15.96–30.98% of total N-terminal residues and 6.81–26.29% of C-terminal residues (Figure 2C). In addition, Pro was the predominant residue at the second N-terminal of other peptides (Figure 2D). It was reported that residues Pro, Leu, and Val at the N-terminal and C-terminal were critical features of highly active DPP-IV inhibitory peptides [44] and residue Pro at the second N-terminal position allows peptides to inhibit the DPP-IV in a competitive manner [46]. Combining knowledge of the structural characterization of DPP-IV inhibitory peptides (i.e., P at the N2 position) and quantitative structure–activity type modeling (QSAR), two peptides with potent in vitro DPP-IV inhibitory activity, LPVPQ (IC_50_ = 43.8 ± 8.8 µM) and IPM (IC_50_ = 69.5 ± 8.7 µM), were reported to have been identified in the intestinal tract following cow milk ingestion [39]. Therefore, these results indicated the presence of highly active DPP-IV inhibitory peptides among the identified peptides in the current study. Further research is required for the efficient selection of DPP-IV inhibitory peptides.

### 3.3. Virtual Screening of Potential DPP-IV Inhibitory Peptides

The prediction of potential bioactive peptides was performed using the PeptideRanking website by evaluating the characteristics of the peptides (e.g., the amino acid composition, the amino acid sequence order, and charge distribution of peptides) [47]. The 698 identified peptides were first predicted for bioactivity with a threshold of 0.5 using the Peptide Ranker website and were then predicted for toxicity using the toxicity website. A total of 240 peptides with a predicted bioactive score greater than 0.5 were considered biologically active, while none of them were predicted to be toxic. Then, 240 potential bioactive peptides were further screened for potential DPP-IV inhibitory peptides through the iDPPIV-SCM tool, LibDock, and sequence features of DPP-IV inhibitory peptides.

The results showed that 215 (89.58%), 223 (92.91%), and 122 (50.83%) of the 240 potential bioactive peptides were predicted to have DPP-IV-inhibitory activity evaluated by the iDPPIV-SCM, LibDock, and sequence features analyses, respectively (Figure 3A). Meanwhile, 200 peptides were consistently predicted by iDPPIV-SCM and LibDock; and the 113 peptides predicted by iDPPIV-SCM and sequence feature analyses, the 114 peptides predicted by LibDock and sequence feature analyses, and the 105 peptides predicted by these three methods are shown in Figure 3A. It was reported that iDPPIV-SCM was a simple, interpretable, and valid tool for identifying and analyzing DPP-IV inhibitory peptides, which could provide accuracies of 0.819 and 0.797 for cross-validation and independent datasets, respectively [48]. The LibDock program could quickly screen target active molecules from complex data based on interactions between ligands and receptors [49]. Compared with traditional methods, these virtual screening approaches could take full advantage of the existing knowledge about peptides, such as protein sequences, cleavage sites of enzymes or chemicals, and ligand–receptor interactions [50], to provide more in-depth studies for screening DPP-IV inhibitory peptides, as well as for probing and potentiating DPP-IV inhibitory activity [48]. These results further confirmed that goat milk was a good source for the generation of DPP-IV inhibitory peptides. It has been reported that camel-milk-derived protein hydrolysates were screened for potential DPP-IV inhibitory peptides based on DPP-IV inhibitory peptide characteristics and the QSAR model. Two of the best DPP-IV inhibitory peptides, LPVP (IC_50_ = 87.0 ± 3.2 µM) and MPVQA (IC_50_ = 93.3 ± 8.0 µM), were identified by the in vitro inhibition test. In addition, the difference in DPP-IV inhibition between camel and cow milk trypsin hydrolysates was considered as due to the different homology between camel and cow milk proteins [51].

Then, the 105 potential DPP-IV inhibitory peptides consensually screened by iDPPIV-SCM, LibDock, and sequence characteristic analyses were further investigated. The distribution of enzymatic hydrolysis sources, molecular weight, and peptide identification scores (PISs) of the 105 peptides is shown in Figure 3B. Among them, about 42.86% of peptides were derived from the enzymatic hydrolysis of whey protein, 18.10% of peptides were derived from casein, and the remaining 39.05% of peptides were present in both (Figure 3B). These results suggested that, in goat milk, the number of DPP-IV inhibitory peptides derived from whey proteins was higher than those from casein proteins. Similar results were reported by Lacroix and Li-Chan [52], who found that peptic hydrolysate of whey proteins had a high inhibitory activity of DPP-IV. On the contrary, these results are inconsistent with the findings from camel and donkey milk, which indicated that digested whey fractions had a lower DPP-IV inhibitory activity compared to the digested casein fractions [38,53]. These discrepancies might be due to the differences in bioactive peptide content of milk from different species, and the content of DPP-IV inhibitory peptides in whey and casein proteins of goat milk were higher than those of donkey and camel milk [54]. In addition, all of the 105 peptides had a molecular weight of less than 1000 Dalton and about 86.67% of the peptides were less than 500 Dalton. These results are consistent with Mu et al. [44], which found that the molecular weights of the obtained DPP-IV inhibitory peptides were almost all between 200 Da and 1000 Da. However, DPP-IV inhibitory peptides with molecular weights greater than 1000 Da, such as PACGGFYISGRPG (IC_50_ = 9.4 μM, 1281.46 Da) and INNQFLPYPY (IC_50_ = 40.08 μM, 1268.43 Da), also showed strong inhibitory activity. Therefore, more studies are needed to further confirm the association between the length of peptides and their inhibitory ability.

### 3.4. Prediction of Pharmacokinetic Properties and Physicochemical Properties

To select the peptide with a relatively high absorption property, the pharmacokinetic properties and physicochemical properties of those 105 peptides were further investigated by ADME evaluation [33] (Appendix A). IPI was used as a positive control in this study [35]. After excluding the peptides that were difficult for intestinal absorption, 18 candidate DPP-IV inhibitory peptides were selected (Table 1). The molecular weight analysis indicated that all 18 peptides have molecular weights of less than 500 Daltons and none of them are derived from casein hydrolysates alone (Figure 3C), which suggested that whey proteins may be superior to casein proteins as the resources of DPP-IV inhibitory peptides in goat milk. Among all of the 18 peptides, except 4 peptides (PLPP, LPPL, PALF, and WVL), all the other peptides had a relatively high PISs (ranging from 24 to 56) (Figure 3C). Because the PISs represented the results of the LC-MS/MS characterization, the results above indicated the high quality of the data [55]. After a systematic literature search, only WVL has been previously reported to have DPP-IV inhibitory activity [56]. The 17 potentially novel DPP-IV inhibitory peptides were used for subsequent analysis. Because 5 peptides (P13, P15-18) were poorly soluble in Silicos-IT class predictions (Table 1) and were predicted to exert potential hepatotoxicity [57], the in vitro DPP-IV inhibition experiments were conducted on the remaining 12 peptides.

### 3.5. Validation and Interactive Mechanisms of Screened DPP-IV Inhibitory Peptides

The DPP-IV inhibitory activities of the 12 potential DPP-IV inhibitory candidates and IPI (P19) are shown in Table 2. IPI is a known strong food-derived DPP-IV inhibitory peptide [40], and its IC_50_ value was found to be 9.28 ± 0.52 μM in this study. Five out of the twelve candidate peptides (VPPF, LPPL, YPF, PALF, LLLP) had detectable DPP-IV inhibitory activities, with an IC_50_ value ranging from 213.99 ± 0.64 μM to 2558.90 ± 5.70 μM (Table 2). Among them, YPF (P4) and LLLP (P9) (mass spectrogram is shown in Appendix A) had relatively potent DPP-IV inhibitory activities (IC_50_ < 500 μM) with IC_50_ values of 368.54 ± 12.97 μM and 213.99 ± 0.64 μM, respectively. These results can be explained in part by the structural and sequential characteristics of these peptides. DPP-IV is a serine protease that principally cleaves peptides containing Pro or Ala at the 2nd N-terminal [43], and the presence of Pro at the 2nd position of the N-terminal in YPF (P4) may facilitate its cleavage by DPP-IV. This was also evidenced by the strong DPP-IV inhibitory ability of IPAVF identified from bovine β-lactoglobulin (IC_50_ = 44.7 μM) [58]. Moreover, YPF (P4) contained amino acids with bulky side chains at the N-terminal, which could provide an enhanced chemical stability and high inhibitory potency of DPP-IV [59]. It was shown that LLLP (P9) contained Leu at the N-terminal and Pro at the C-terminal, which had been reported to be important characteristics of potential DPP-IV inhibitory peptides [44]. These results are in agreement with the findings of Nongonierma et al. [60] which showed that tri- and tetrapeptides containing Pro at the C-terminus were DPP-IV inhibitors, and FLEP (IC_50_ = 65.3 ± 3.5 μM) was the most potent inhibitor. This finding was also reported by Harnedy et al. [61] who found that tetrapeptide LLAP had high DPP-IV inhibitory activity. These results suggested that LL at the N-terminal may be connected with a high DPP-IV inhibitory activity [43]. However, LLW (P14) in our study showed no DPP-IV inhibitory activity. These results are in agreement with those obtained by Tulipano et al. [62], which found that the dipeptides LL showed DPP-IV inhibitory activity. However, the inhibitory activity of LLF was the opposite. These results indicated the importance of residues present at the C-terminal on the DPP-IV inhibitory activity of the peptides [43].

In addition, YPPF (P2) and PALF (P5) had general DPP-IV inhibitory activities with IC_50_ values of 1573.85 ± 256.05 μM and 1746.7 ± 29.90 μM, respectively. P2 and P4 have similar sequences but very different inhibitory activities, which may be due to the presence of Pro at the third N-terminal of P2. Similarly, the presence of Pro at the third N-terminal of PLPP (P1, ND), LPPL (P3, IC_50_ = 2558.90 ± 5.70 μM), and LLPL (P8, ND) also resulted in poor inhibitory activity. This observation was similar to other studies that peptides containing Pro, hydroxyproline, or *N*-methyl glycine at the third position of the N-terminal were not cleaved by DPP-IV [43], and the first two peptide bonds at the N-terminal must be in a trans-configuration [45]. Although P5 contained Ala at the second N-terminus, which is characteristic of DPP-IV substrates, the inhibitory activity of P5 was not very strong. Lacroix and Li-Chan [14] reported that the binding affinity of peptides with DPP-IV was related to the type of amino acid residue at the second N-terminal of peptides, and the affinity greatly reduced in the order of Pro, Ala, Gly, hydroxyproline, and other small uncharged residues. This study showed that peptides containing Pro at the 2nd N-terminal were more easily cleaved by DPP-IV, while peptides containing Ala at 2nd N-terminal were hydrolyzed at a much lower rate [63]. Therefore, this may be the reason why the inhibitory activity of P5 is much lower than that of P4.

It was surprising that the inhibitory activities of the five peptides, FVY (P6), LYL (P7), LVW (P11), LWV (P12), and P14, were not detectable despite the presence of DPP-IV inhibitory peptide characteristic amino acids, such as Leu, Phe, and Val at the N-terminal [44]. Nongonierma et al. [64] also found that peptides derived from milk proteins containing hydrophobic or aromatic amino acids at the N-terminus had no DPP-IV inhibitory activity and concluded that this feature was not sufficient to generate inhibitory activity. These results indicated that the presence of amino acids at the N-terminal of DPP-IV inhibitory peptides may play an auxiliary role and more studies are needed to further explore the structure–effect relationships of DPP-IV inhibitory peptides.

### 3.6. Cell Viability and In Situ DPP-IV Inhibition of Screening Peptides on Caco-2 Cells

The cell viability and in situ DPP-IV inhibitory effects of YPF and LLLP on Caco-2 cells were studied according to the results of in vitro inhibitory experiments. As shown in Figure 4A,B, there was no significant decrease in cell viability when YPF and LLLP concentrations were 0.025~1 mg/mL (*p* > 0.05), indicating that YPF and LLLP were nontoxic to Caco-2 cells during 12 h of incubation. Moreover, the inhibition of YPF and LLLP on the DPP-IV activity of Caco-2 cells was determined, and the results obtained are shown in Figure 4C,D. As a positive control, the IC_50_ value of IPI was 10.85 ± 2.72 µM, which is similar to previous reports by You et al. [36] and Harnedy-Rothwell et al. [65]. The results showed that YPF and LLLP possessed good in situ DPP-IV inhibitory activity with IC_50_ values of 159.46 ± 17.40 µM and 154.96 ± 8.41 µM, respectively. Interestingly, the DPP-IV inhibition of YPF and LLLP in cell-based assays was higher than that of in vitro chemical system inhibition assays, suggesting that both were potentially resistant to degradation by membrane-associated peptidases. However, further studies on the ability of YPF and LLLP to pass through the cell membrane intact are needed in the future.

### 3.7. Interactive Mechanism of Screening Peptides against DPP-IV

Then, we examined the possible inhibitory mechanisms of the novel peptides (YPF and LLLP) against DPP-IV (5J3J, B chain) at the molecular level. The reference ligand HL1 was used as the control, and the potential mechanism is illustrated in Figure 5. The results revealed that peptides YPF (P4) (56.22 kcal/mol) and LLLP (P9) (62.95 kcal/mol) bound on the active pocket of 5J3J (B chain) with higher CDOCKER interaction energy scores than HL1(48.75 kcal/mol) (Figure 5) and the non-bonding interaction modes are drawn in forms of both 3D and 2D diagrams. These results indicated that hydrogen bonds and hydrophobic interactions were the dominant binding modes of P4/P9 with 5J3J (B chain), including 9/4 hydrogen bonds and 4/11 hydrophobic interactions, respectively (Figure 5F). In contrast, 9 hydrogen bonds and 5 π-participated hydrophobic interactions were the binding modes of reference ligand HL1 with 5J3J (B chain) formed (Figure 5F). Previous studies indicated that the hydrophobic S1 pocket was composed of amino acid residues TYR631, VAL656, TRP659, TYR662, TYR666, and VAL711 [66], while the charged S2 pocket was composed of ARG125, GLU205, GLU206, SER209, PHE357, and ARG358 [67]. S1 and S2 together constituted the active site of DPP-IV (Figure 5D). The results of this study indicated that P4 bound to DPP-IV through binding to active residues TYR666 and TYR662 in the S1 pocket and to active residues ARG358, GLU205, GLU206, and PHE357 in the S2 pocket; P9 bound to DPP-IV through interactions with active residues TYR631, TYR666, VAL656, TYR662, and VAL711 in the S1 pocket and with active residues ARG125, SER209, ARG358, PHE357, and GLU205 in the S2 pocket. These differences might partially account for the different DPP-IV inhibitory capacity of those three ligands above. These results are supported by the previous findings that peptides IPYWTY (IC_50_ = 11.04 μΜ) and IPYWT (IC_50_ = 18.29 μΜ) had high inhibitory activities due to their hydrophobic interaction and hydrogen bond with the S1 pocket of DPP-IV [24]. Moreover, it was reported that the S1 pocket plays a key role in inhibiting DPP-IV since the pyrrolidine ring, a potent DPP-IV inhibitor, can bind to the S1 pocket by Val711, Tyr662, Trp659, Tyr666, and val656 residues [68]. Overall, it is suggested that the two identified peptides, especially P9, could better bind to DPP-IV active pockets and exert their DPP-IV inhibitory activities.

Through statistical analysis of the non-binding interaction sites, we further understood the molecular mechanism of peptides (YPF and LLLP) against DPP-IV and the results are shown in Figure 5E. The results showed that the two clusters covered all interaction sites. The composition of four residues of the S2 pocket of DPP-IV constituted Cluster 1, including ARG358, PHE357, TYR662, and GLU205 [66], and they were mainly involved in the interaction with the ligand by forming hydrogen bonds and electrostatic attraction (Figure 5A–C). These four amino acid residues were involved in the P4, P9, and HL1 non-bonding interaction. Cluster 2 consisted of the S1 pocket, S2 pocket, and other interaction sites outside the active cavity of DPP-IV [67] and exhibited different interaction patterns with different ligands (Figure 5A–C). In addition, the ligands could also be classified into two groups, of which P4 forms 9 hydrogen bonds (GLU205, GLU206, ARG358, TYR666, TYR547, SER552, and TYR662), 4 hydrophobic interactions (PHE357, TYR662, TYR547, and TYR666), and an electrostatic attraction (GLU205) with DPP-IV (Figure 5A,F). The residues interacting with P4 mainly belong to the S2 pocket, which was also the reason why they had fewer hydrophobic interactions. The other group includes P9 and HL1, which interacted more with amino acid residues in the S1 pocket than P4. Figure 5B,F show that P9 forms 5 hydrogen bonds (ARG125, SER209, GLU205, TYR662, and ARG358), 11 hydrophobic interactions (HIS126, TYR631, TYR547, PHE357, TYR662, TYR666, VAL656, and VAL711), 3 electrostatic attraction interactions (GLU205, PHE357, and ARG358) and Unfavorable Donor–Donor interactions (ASN710) with DPP-IV. Therefore, the 11 hydrophobic interactions between P9 and amino acid residues in active pockets may be the reason why the DPP-IV inhibitory activity of P9 was significantly higher than that of P4. Overall, the inhibitory abilities of YPF and LLLP against DPP-IV can be mainly attributed to the formation of strong non-bonding interactions with the core residues ARG358, PHE357, GLU205, TYR662, TYR547, and TYR666 from active pockets of DPP-IV (Figure 5E). These results indicated that P4 and P9 were able to bind in a reasonable conformation to the active centers consisting of the two pockets S1 and S2 of DPP-IV, thereby inhibiting the activity of DPP-IV.

## 4. Conclusions

In the present study, two novel DPP-IV inhibitory peptides (YPF and LLLP) were screened and identified from goat milk hydrolysate by peptidomics and the in silico screening method. The identified novel peptides exhibited relatively high DPP-IV inhibitory activity both in the in vitro chemical system (YPF, IC_50_ = 368.54 ± 12.97 μM; and LLLP, IC_50_ = 213.99 ± 0.64 μM) and in situ (YPF, IC_50_ = 159.46 ± 17.40 μM; and LLLP, IC_50_ = 154.96 ± 8.41 μM). The potential inhibitory mechanisms of the DPP-IV inhibitory peptides (YPF and LLLP) might primarily be attributed to the formation of strong non-bonding interactions with the core residues ARG358, PHE357, GLU205, TYR662, TYR547, and TYR666 from active pockets of DPP-IV. Overall, this study provided theoretical support for the rapid screening of DPP-IV inhibitory peptides from goat milk as well as for the study of other homologous milks, and provided insight into the intrinsic mechanism of the DPP-IV inhibitory activity of the obtained peptides.

## Figures and Tables

**Figure 1 foods-13-01194-f001:**
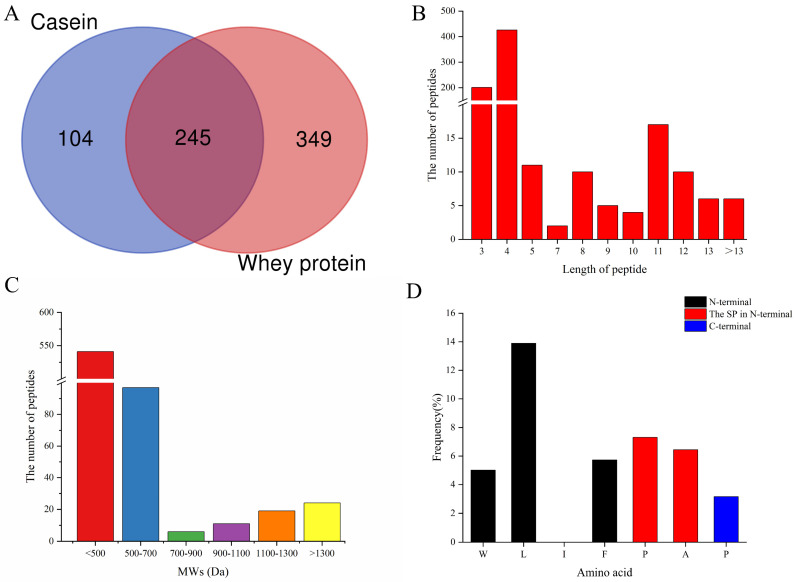
General description of 698 peptides identified from enzymatic hydrolysis products of goat milk whey protein and casein by LC-MS/MS. (**A**) The source distribution of 698 peptides identified. (**B**) The length distribution of 698 identified peptides. (**C**) Molecular weight distributions of 698 peptides were identified. (**D**) The distribution of amino acid residues (Trp, Leu, Ile, or Phe) at the N-terminus, and/or a Pro/Ala at position 2, and/or Pro at the C-terminus of 698 identified peptides.

**Figure 2 foods-13-01194-f002:**
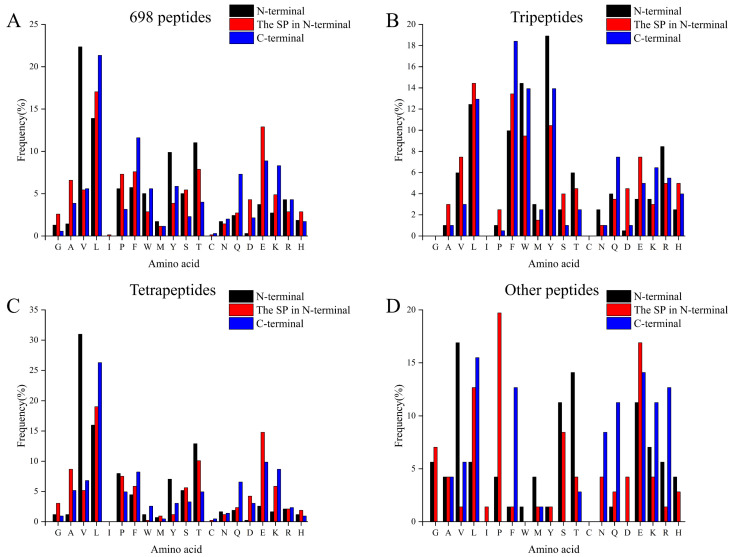
The distribution of 20 amino acids at the N-terminal/2nd N-terminal/C-terminal. (**A**) The distribution of 20 amino acids in the N-terminal/2nd N-terminal/C-terminal of 698 identified peptides. (**B**) The distribution of 20 amino acids on the N-terminal/2nd N-terminal/C-terminal of the identified tripeptide. (**C**) The distribution of 20 amino acids on the N-terminal/2nd N-terminal/C-terminal of the identified tetrapeptides. (**D**) The distribution of 20 amino acids on the N-terminal/2nd N-terminal/C-terminal of identified peptides larger than four amino acid residues.

**Figure 3 foods-13-01194-f003:**
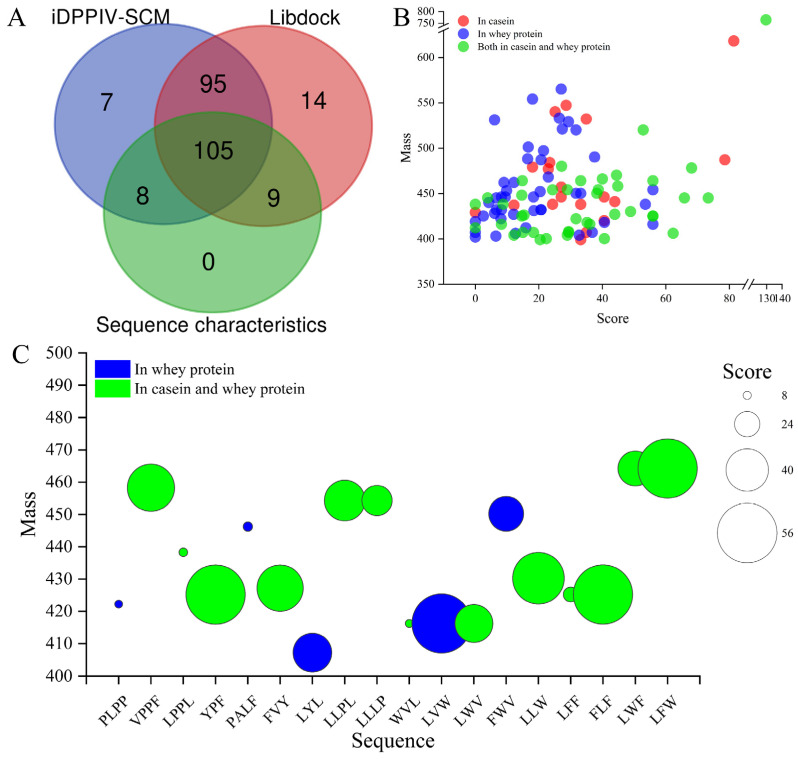
The 18 potential DPP-IV inhibitory peptides were virtually screened from 240 active peptides. (**A**) Potential DPP-IV inhibitory peptides were virtually screened by iDPPIV-SCM, Libdock, and sequence characteristics of DPP-IV inhibitory peptides. (**B**) Molecular weight, peptide identification score by LC-MS/MS, and source distribution of enzymatic hydrolysis of 105 potential DPP-IV inhibitory peptides. (**C**) Molecular weight, sequence, and source distribution of potential DPP-IV inhibitory peptides with high GI for 18 of the 105 peptides predicted using the SwissADME website.

**Figure 4 foods-13-01194-f004:**
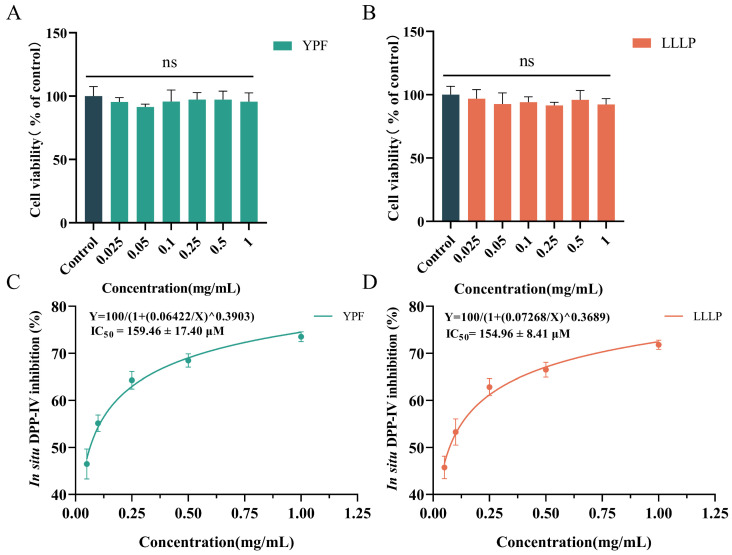
Cell viability and in situ DPP-IV inhibition of screening peptides on Caco-2 cells. (**A**,**B**) The cell viability of YPF and LLLP in Caco-2 cells for 12 h. (**C**,**D**) In situ DPP-IV inhibitory activities of YPF and LLLP in Caco-2 cells. Data are presented as mean ± SD (*n* = 3). ns indicates that the samples were not significantly different from the control (*p* > 0.05).

**Figure 5 foods-13-01194-f005:**
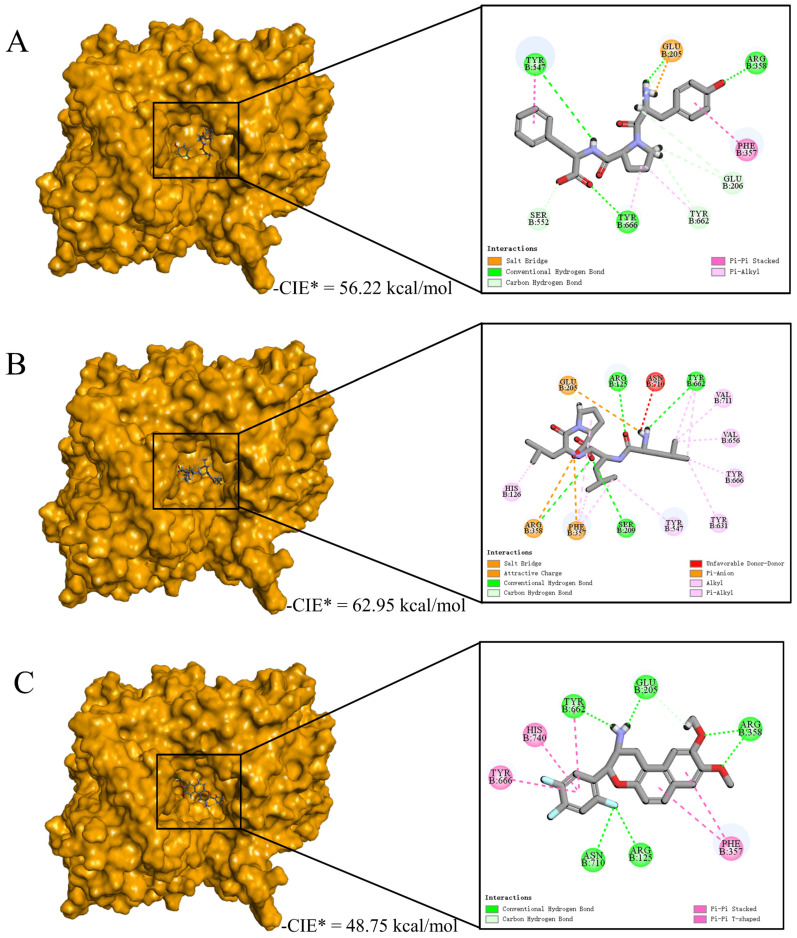
Underlying molecular mechanisms of identified peptides from goat milk against DPP-IV revealed by molecular docking. (**A**−**C**) Three-dimensional and two-dimensional diagrams of P4 (**A**), P9 (**B**), and HL1 (**C**) binding with DPP-IV (B−chain), respectively. (**D**) The 3D diagram of the S1 and S2 active cavity of DPP-IV (B−chain). (**E**) HCA analysis of non-bonding interactive sites between the ligands (YPF, LLLP, and HL1) and DPP-IV. The shade of color indicated the number of non−bond binding interactions with key amino acid residues. (**F**) The statistics of non-bonded interaction types of ligands (YPF, LLLP, and HL1) interacting with DPP-IV.

**Table 1 foods-13-01194-t001:** Major properties of 18 potential DPP-IV inhibitory peptides versus control IPI.

Peptide	Sequence	Water Solubility (ESOL)	Water Solubility (Ali)	Water Solubility (Silicos-IT)	GI Absorption ^a^	BBB Permeant ^b^	P-gp Substrate ^c^	CYP1A2 Inhibitor ^d^	CYP2C19 Inhibitor ^e^	CYP2C9 Inhibitor ^f^	CYP2D6 Inhibitor ^g^	CYP3A4 Inhibitor ^h^
P1	PLPP	VS	VS	S	High	No	Yes	No	No	No	No	No
P2	VPPF	VS	VS	S	High	No	Yes	No	No	No	No	No
P3	LPPL	VS	VS	S	High	No	Yes	No	No	No	No	No
P4	YPF	VS	VS	MS	High	No	Yes	No	No	No	No	No
P5	PALF	VS	VS	MS	High	No	Yes	No	No	No	No	No
P6	FVY	VS	VS	MS	High	No	Yes	No	No	No	No	No
P7	LYL	VS	VS	S	High	No	Yes	No	No	No	No	No
P8	LLPL	VS	VS	S	High	No	Yes	No	No	No	No	No
P9	LLLP	VS	VS	S	High	No	Yes	No	No	No	No	No
P10	WVL	VS	VS	MS	High	No	Yes	No	No	No	No	No
P11	LVW	VS	VS	MS	High	No	Yes	No	No	No	No	No
P12	LWV	VS	VS	MS	High	No	Yes	No	No	No	No	No
P13	FWV	S	S	PS	High	No	Yes	No	No	No	Yes	No
P14	LLW	VS	S	MS	High	No	Yes	No	No	No	No	No
P15	LFF	VS	VS	PS	High	No	Yes	No	No	No	Yes	Yes
P16	FLF	VS	VS	PS	High	No	Yes	No	No	No	Yes	Yes
P17	LWF	S	S	PS	High	No	Yes	No	No	No	Yes	No
P18	LFW	S	S	PS	High	No	Yes	No	No	No	Yes	No
P19	IPI	VS	VS	S	High	No	Yes	No	No	No	No	No

^a^ indicates the prediction of gastrointestinal absorption; ^b^ indicates the prediction of penetration of the blood—brain barrier; ^c^ indicates the prediction of P-glycoprotein substrate inhibitors; ^d–h^ indicate the prediction of inhibitors of different isoforms of cytochromes P450 (CYP). “Very soluble”, “Soluble”, “moderately soluble”, and “Poorly soluble” have been abbreviated and named as “VS”, “S”, “MS”, and “PS”.

**Table 2 foods-13-01194-t002:** Physicochemical properties and DPP-IV inhibitory activity of peptide by in silico analysis.

Peptide	Sequence	Mass (Da)	Retention Time (min)	Identification Score	DPP-IV Inhibitory Activity (IC_50_, μM)	Source
P1	PLPP	422.25	13.884	8.0791	ND	Whey protein
P2	VPPF	458.25	50.601	44.845	1573.85 ± 256.05	Casein and whey protein
P3	LPPL	438.28	27.735	8.7137	2558.90 ± 5.70	Casein and whey protein
P4	YPF	425.19	24.940	55.918	368.54 ± 12.97	Casein and whey protein
P5	PALF	446.25	38.521	9.4209	1746.7 ± 29.90	Whey protein
P6	FVY	427.21	21.367	43.876	ND	Casein and whey protein
P7	LYL	407.24	28.303	36.833	ND	Whey protein
P8	LLPL	454.31	35.932	38.717	ND	Casein and whey protein
P9	LLLP	454.31	32.377	28.962	213.99 ± 0.64	Casein and whey protein
P11	LVW	416.24	30.462	55.918	ND	Whey protein
P12	LWV	416.24	29.518	36.05	ND	Casein and whey protein
P14	LLW	430.26	36.161	48.874	ND	Casein and whey protein
P19	IPI	341.23	-	-	9.28 ± 0.52	-

## Data Availability

The original contributions presented in the study are included in the article/Appendix A, further inquiries can be directed to the corresponding authors.

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
