# Peer review of "Mechanistic Study of Novel Dipeptidyl Peptidase IV Inhibitory Peptides from Goat’s Milk Based on Peptidomics and In Silico Analysis"

_foods, 2024, doi:10.3390/foods13081194_

Round 1
Reviewer 1 Report
Comments and Suggestions for Authors
In this study, the objectives were to rapidly identify and screen potential DPP-4 inhibitory peptides from goat milk based on in silico screening, LC-MS/MS, molecular docking, and in vitro assessment. The sequence characteristics and the molecular binding mechanisms of these peptides were also investigated. The study may contribute to rapidly screening DPP-IV inhibitory peptides from goat milk and promote the application of goat milk as functional food.
The main title is too long, it should be shortened.
DPP-4 or DPP-IV, it's confusing.
Goat's milk, for what reasons it was chosen.
Goat's milk as a material was not described at all in this study.
What was the quality, chemical composition, etc., of the used goat's milk? How was it collected?
In the discussion of the results, data on these or similar studies with different types of milk, such as cow's and similar, are missing.
As a summary or conclusion, the possibility of applying the obtained results to nutrition in the case of other types of milk should be added.
Accordingly, the literature used is not complete.
it is not clear what MS Excel files Document 1 and Document 2 represent. The data should be edited differently in these files, so that they are part of the Supplementary material (data).
Author Response
We are very pleased in receiving your letter together with the reviewer's comments for our manuscript "Foods-2920443". We thank the reviewers for thoroughly reviewing our manuscript and making thoughtful comments. The manuscript has been revised based on the reviewers’ comments and suggestions, and the details about the revision were highlighted in the revised manuscript and listed as follow point by point.
- The main title is too long, it should be shortened.
Response: We appreciated reviewer’s useful comments. As requested, we have changed the title to 《 Mechanistic study of novel dipeptidyl peptidase IV inhibitory peptides from goat's milk based on peptidomics and in silico analysis 》 and highlighted it in the manuscript.
- DPP-4 or DPP-IV, it's confusing.
Response: Thanks so much for the valuable suggestion. We've standardized on "DPP-IV" in the manuscript.
- Goat's milk, for what reasons it was chosen.
Response: Thanks for the good question. Goat milk is widely consumed around the world for its extensive nutritional properties that make it exceptional and maintain the health of children and adults [1]. Bioactive peptides derived from goat milk exert multifunctional properties, including anti-microbial, immunomodulatory, cholesterol-lowering, anti-oxidant, anti-thrombotic, antagonistic activities against various toxic agents [2], as well the inhibitory activities on DPP-IV [3]. Novel DPP-IV inhibitory peptides were continuously identified by different 64 studies [4], highlighting the immense value of goat milk in producing food protein-derived DPP-IV inhibitor. It has been reported that compared with cow milk, goat milk has higher digestibility, lower sensitization and health promoting benefits [5]. (Lin59-67)
References:
[1] Jankiewicz, M.; van Lee, L.; Biesheuvel, M.; Brouwer-Brolsma, E.M.; van der Zee, L.; Szajewska, H. The Effect of Goat-Milk-Based Infant Formulas on Growth and Safety Parameters: A Systematic Review and Meta-Analysis. Nutrients. 2023, 15. https://doi:10.3390/nu15092110.
[2] Mohanty, D.P.; Mohapatra, S.; Misra, S.; Sahu, P.S. Milk Derived Bioactive Peptides and Their Impact on Human Health – A Review. Saudi J Biol Sci. 2016, 23, 577–583. https://doi: 10.1016/j.sjbs.2015.06.005.
[3] Zhang, Y.; Chen, R.; Ma, H.; Chen, S. Isolation and Identification of Dipeptidyl Peptidase IV-Inhibitory Peptides from Tryp-sin/Chymotrypsin-Treated Goat Milk Casein Hydrolysates by 2D-TLC and LC-MS/MS. J Agric Food Chem. 2015, 63, 8819–8828. https://doi: 10.1021/acs.jafc.5b03062.
[4] Du, X.; Jiang, C.; Wang, S.; Jing, H.; Mo, L.; Ma, C.; Wang, H. Preparation, Identification, and Inhibitory Mechanism of Di-peptidyl Peptidase IV Inhibitory Peptides from Goat Milk Whey Protein. J Food Sci. 2023, 88, 3577–3593. https://doi:10.1111/1750-3841.16694.
[5] Dos Santos, W.M.; Guimarães Gomes, A.C.; de Caldas Nobre, M.S.; de Souza Pereira, Á.M.; dos Santos Pereira, E.V.; dos Santos, K.M.O.; Florentino, E.R.; Alonso Buriti, F.C. Goat Milk as a Natural Source of Bioactive Compounds and Strategies to Enhance the Amount of These Beneficial Components. Int Dairy J. 2023, 137. https://doi: 10.1016/j.idairyj.2022.105515.
- Goat's milk as a material was not described at all in this study.
Response: Thanks for the good question. We have added goat's milk as an experimental material in the materials section.
- What was the quality, chemical composition, etc., of the used goat's milk? How was it collected?
Response: Thank you very much for your comments. In fact, the goat milk used for our experiments was selected from the goat milk of Sanen, Weinan City, Shaanxi Province, with a protein content of 3.6%, a fat content of 5.8%, a lactose content of 4.6%, and a mineral content of 0.86%. Goat milk collected between September and October 2021 was transported by ice packs under aseptic conditions and then stored at -80 ◦C until use (Lin-87-90).
- In the discussion of the results, data on these or similar studies with different types of milk, such as cow's and similar, are missing.
Response: Thanks for your great suggestion. We have added data from relevant research data on different types of milk to the discussion of results (Line271-272, 283-287,332-336, 366-372, 439-440, 476-478). Additionally, it was shown that the amino acid sequences of goat and bovine β-casein have a high degree of homology (91%) [1], so it could be assumed that goat milk was homologous to cow milk protein. In addition, homologous proteins have the potential to produce peptides with similar biological functions [2], so cow milk also has the potential to produce large amounts of DPP-IV inhibitory peptides. The above statement has been added to the corresponding section of the manuscript (Line281-285).
References:
[1] Wu, Y.; Zhang, J.; Mu, T.; Zhang, H.; Cao, J.; Li, H.; Tang, H.; Chen, L.; Liu, H.; Xu, X.; et al. Selection of Goat β-Casein Derived ACE-Inhibitory Peptide SQPK and Insights into Its Effect and Regulatory Mechanism on the Function of Endothelial Cells. Int J Biol Macromol. 2023, 253. https://doi: 10.1016/j.ijbiomac.2023.127312.
[2] Zhang, J.; Wu, Y.; Tang, H.; Li, H.; Da, S.; Ciren, D.; Peng, X.; Zhao, K. Identification, Characterization, and Insights into the Mechanism of Novel Dipeptidyl Peptidase-IV Inhibitory Peptides from Yak Hemoglobin by in Silico Exploration, Molecular Docking, and in Vitro Assessment. Int J Biol Macromol. 2024, 259. https://doi: 10.1016/j.ijbiomac.2023.129191.
- As a summary or conclusion, the possibility of applying the obtained results to nutrition in the case of other types of milk should be added.
Response: Thanks so much for the valuable suggestion. We modified the conclusions of the study by adding the possibility of applying the obtained results to other types of milk in the manuscript (Lin566-569). The following statement: Overall, this study provided theoretical support for the rapid screening of DPP-IV inhibitory peptides from goat milk as well as for the study of other homologous milk, and provided insight into the intrinsic mechanism of the DPP-IV inhibitory activity of the obtained peptides.
- Accordingly, the literature used is not complete.
Response: Thanks for the suggestion. As requested, we have added to the corresponding sections of the manuscript and cited relevant literature to illustrate our study.
- It is not clear what MS Excel files Document 1 and Document 2 represent. The data should be edited differently in these files, so that they are part of the Supplementary material (data).
Response: Thanks so much for the valuable suggestion. The contents of Document 1 and Document 2 have been edited and dissolved in the Supplementary File to appear in the manuscript as Table S1 and Table S2, respectively. (Lin265, Line393)
Reviewer 2 Report
Comments and Suggestions for Authors
The manuscript is interesting, novel and addresses highly topical issues. In general, the methods applied are well documented, although perhaps they have been described in an excessively dense way to facilitate reading. Only a few minor aspects relating more to the format than to the content can be criticized:
-Abstract section: "overall....functional food". This phrase is perfectly dispensable since only the peptides have been isolated and their activity has been tested, but considering a food as "functional" is more complex.
-Quality of Fgure 1 is low, and poorer than the other figures. Please improve its resolution.
-Text in lines 294-304 was not justified
-Table 1 size and and its reading could be improved if the terms "Very soluble", "Soluble" and "moderately soluble" are abbreviated and named as "S", "VS" or "MS".
-References format are not acording to the Foods´guidelines for authors, especially in the journal names and year of publication. Please follow Foods´ intructions for authors.
Author Response
We are very pleased in receiving your letter together with the reviewer's comments for our manuscript "Foods-2920443". We thank the reviewers for thoroughly reviewing our manuscript and making thoughtful comments. The manuscript has been revised based on the reviewers’ comments and suggestions, and the details about the revision were highlighted in the revised manuscript and listed as follow point by point.
- Abstract section: "overall....functional food". This phrase is perfectly dispensable since only the peptides have been isolated and their activity has been tested, but considering a food as "functional" is more complex.
Response: Thanks so much for the valuable suggestion. We have revised the relevant sections in the manuscript as follows: Overall, the two novel DPP-IV inhibitory peptides rapidly identified in this study can be used as functional food ingredients for the control of diabetes (Lin229-31).
- Quality of Fgure 1 is low, and poorer than the other figures. Please improve its resolution.
Response: Thanks for the suggestion. Revised as suggested.
- Text in lines 294-304 was not justified
Response: Really sorry for the mistake. Revised as suggested.
- Table 1 size and and its reading could be improved if the terms "Very soluble", "Soluble" and "moderately soluble" are abbreviated and named as "S", "VS" or "MS".
Response: We really appreciate your valuable comments. We have made changes and highlighted them in the manuscript as suggested (Line408-413).
- References format are not acording to the Foods´guidelines for authors, especially in the journal names and year of publication. Please follow Foods´ intructions for authors.
Response: Thanks so much for your valuable suggestion. We have revised and highlighted all references in accordance with foods magazine's reference requirements.
Round 2
Reviewer 1 Report
Comments and Suggestions for Authors
The authors "item by item" corrected the previous version of the manuscript and significantly improved it.